# Assessing the Polarising Impacts of Low-Traffic Neighbourhoods: A Community Perspective from Birmingham, UK

**DOI:** 10.3390/ijerph21121638

**Published:** 2024-12-09

**Authors:** Isabelle Whelan, Carlo Luiu, Francis D. Pope

**Affiliations:** School of Geography, Earth and Environmental Sciences, University of Birmingham, Birmingham B15 2TT, UK; ixw174@student.bham.ac.uk (I.W.); c.luiu@bham.ac.uk (C.L.)

**Keywords:** low-traffic neighbourhood, low-emission zones, modal filter, air pollution, active travel, motorised transport, non-motorised transport, polarisation

## Abstract

Globally, the transport sector is a major contributor to air pollution. Currently, in the UK, vehicle emissions contribute significant amounts of nitrogen oxide (NOx) and particulate matter (PM) pollution in urban areas. Low-emission-zone policies have been used as an intervention to tackle air pollution, and in this context, the UK launched the Low-Traffic Neighbourhood scheme. This study investigates the impacts of the Low-Traffic Neighbourhood in Kings Heath, Birmingham, UK, to evaluate its impact in reducing air pollution and local community perspectives about the scheme and perceived impacts on health and well-being. This study employs a mixed-method approach comprising an air-quality-monitoring assessment and a survey questionnaire involving 210 residents. The findings reveal an increase in active travel and a reduction in air pollution levels in the years after the implementation of the scheme, although the area is still non-compliant with the 2021 WHO air quality guidelines. Nonetheless, the scheme has a polarising effect and created a division within the local community about the overall scheme acceptance and spatial distribution of the scheme’s benefits. This study underscores the importance of comprehensive baseline data, long-term community engagement, and integration with broader urban planning initiatives to enhance the success of future Low-Neighbourhood Traffic schemes.

## 1. Introduction

### 1.1. Health Impact of Air Pollution Impacts

Air pollution is recognised as the contamination of the atmosphere by any chemical, physical, or biological agent at increased concentrations, emitted from natural or anthropogenic sources, posing risks to human health [1,2]. The World Health Organisation (WHO) identifies air pollution as the foremost environmental health threat, second only to tobacco smoking as the leading cause of non-communicable diseases and mortality in Europe [1,3]. Air pollution contributes to one in six deaths globally, responsible for approximately 9 million deaths per year [4]. In the UK, air pollution causes approximately 28,000 to 36,000 deaths annually [5]. The detrimental health impacts of air pollution are rarely contested [2], and in 2013, Ella Kissi-Debrah (9 years old) was the first UK case where air pollution was listed as a cause of death [5]. In 2014, ambient air pollution and particulate matter (PM) were classified as Group 1 carcinogens for humans, with acute exposures to high concentrations of nitrogen dioxide (NO_2_) and PM inducing oxidative stress and inflammation and increasing morbidity, mortality, and emergency hospital admissions for cardiovascular diseases (CVDs), lung cancer, respiratory conditions, cerebrovascular diseases, and ischemic heart disease, the leading cause of death in the UK [6,7,8]. Vulnerable groups are particularly affected with an impact extending beyond health, with estimated costs to the NHS due to air pollutants at £1.6bn between 2017 and 2025 and a loss of economic productivity from road congestion [9].

Since 1987, the WHO has issued air quality guidelines (AQGs), setting recommended levels and interim targets for common air pollutants and highlighting clean air as a basic human right [10]. In order to better address the threats caused by air pollution, in 2021, the WHO revised its 2005 Global AQGs [11]. The modification of the AQGs was prompted by a growing body of evidence illustrating the detrimental effects of air pollution at levels lower than previously recognised, with pollution level adjustment to increase the safeguard of public health and reduce the worldwide disease burden linked to air pollution. The revised guidelines establish more stringent restrictions on pollutants such as PM_2.5_, PM_10_, and NO_2_, stressing the need for enhanced regulations to alleviate the health consequences of air pollution [10,12] (see Table 1). In the UK, air pollution limits are regulated by the Air Quality Standards (AQS) Regulations 2010, which are part of the Air Quality Strategy [13]. As Table 1 shows, the UK AQSs are generally less stringent than the WHO guidelines, particularly for PM_2.5_, PM_10_, and NO_2_.

### 1.2. Low-Traffic Neighbourhoods as a Potential Solution to Tackle Air Pollution

The transport sector is one of the main contributors to environmental deterioration, contributing to nearly a quarter of energy-related carbon emissions globally [14]. In the UK, 27% of domestic greenhouse gas emissions and air pollution are attributable to the transport sector [15]. In 2019, more than 80% of England’s population resided in urban areas [16], where population density increased vehicular emissions and significantly depleted air quality [17,18,19]. Despite global efforts to reduce vehicular emissions through stricter emission standards, NO_2_ and PM persist above the WHO AQGs, and the growth of urban populations has introduced greater congestion levels, offsetting some benefits [20,21,22]. Urban air pollution disproportionately affects urban communities due to their proximity to traffic congestion [2,23]. Vehicle emissions, primarily from internal combustion engines burning fossil fuels, contribute up to two-thirds of urban pollution, with transport accounting for 32% of NOx emissions and 14% of PM_2.5_ [24], with diesel engines emitting higher levels of NOx and PM compared to petrol [25].

As part of the strategy to tackle road-traffic-related air pollution in urban contexts, low-emission zones (LEZs) have been developed and implemented widely in Europe [26,27]. LEZs consist of urban areas that can only be accessed by vehicles that meet specific emissions standards (e.g., Euro standards), prohibiting or disincentivizing older and more polluting vehicles from operating [28]. In the UK, LEZs, also known as ultra-low-emission zones or clean air zones, are an integral part of the air quality [13] and transport decarbonisation [15] strategies. As a further intervention to encourage active travel and physical activity, reducing car usage, and reaping potential benefits from reductions in air pollution, noise, and road-related casualties, the UK Department for Transport launched low traffic neighbourhoods (LTNs) in 2020. LTNs can be defined as “a traffic management scheme aimed at reducing or removing through traffic from residential areas, put in place using traffic signed restrictions or physical measures such as planters or bollards” [29]. LTNs are small-scale area-based interventions employing modal filters like planters, bollards, or camera gates to limit through-traffic and control vehicle speeds on residential streets [30,31].

LTNs have emerged as a strategy to address traffic congestion, air pollution, and the promotion of active travel. To date, the majority of evidence related to LTNs stems from research surrounding their implementation in London [32]. Existing research highlights the potential of well-designed LTNs to enhance urban living by fostering active transportation, easing congestion, and improving air quality. Laverty et al. [33] and Xiao et al. [34] suggest that LTNs employ “carrots” (positive incentives) to create safer walking and cycling environments and “sticks” (negative incentives) to discourage convenient car journeys. Brand et al. [35] stressed that LTNs expand the space for active transportation without reducing car access, while Haigh [36] emphasised the need to combine both approaches to effectively limit car usage. Yang et al. [31] found that LTNs can reduce air pollution and traffic within designated areas without adverse effects on neighbouring streets. Implementation led to significant reductions in NO_2_ levels within intervention areas (5.7%) and boundary areas (8.9%), alongside substantial traffic volume reductions within LTNs (58.2%). Similarly, Thomas and Aldred [37] observed reduced traffic volumes and driving behaviours across all LTN zones in London. In addition to a decrease in traffic and associated air pollution, Goodman and Aldred [38] observed a 10% decrease in total street crime following LTN implementation. Xiao et al. [34] suggest that any enhanced perceived safety may promote active travel, benefiting vulnerable groups [39]. Goodman et al. [40] supports this finding by observing a seven-fold increase in school children cycling near LTNs in Dulwich Village, highlighting safety improvements. While Goodman and Aldred [38] found decreases in crime rates and LTN implementation, concerns about hindered emergency response times due to reduced vehicle access are mitigated by Goodman et al. [41], who found no adverse effects on emergency response times.

Nonetheless, despite their intended benefits, LTNs often face controversy within communities [31,42]. Community controversy persisted as boundary road residents believed benefits were localised, as evidenced by Xiao et al. [34], who noted decreased daily traffic counts within LTNs but remained unchanged on boundary streets. Similarly, Pritchett et al. [32] found that in Kings Heath LTN in Birmingham, residents reported, in the post-implementation consultations, negative effects of increased congestion outside the LTN boundaries, negative and divisive effects of modal filters on traffic in the neighbourhood, experiencing inconvenient journeys that necessitate car use, increased high street traffic and longer journeys to the high street, and negative effects on pollution and safety on roads without modal filters.

### 1.3. Kings Heath Low-Traffic Neighbourhood Zone

Birmingham City Council (BCC) declared itself an air quality management area for NO_2_ in 2010 [43]. Enhanced emission monitoring since 2018 supports the ‘Road to Zero’ strategy, aiming for petrol and diesel car sales to end by 2040. Jafari et al. [44] state that local actions like LEZs, clean air zones, and LTNs complement EU directives and the convention on long-range transboundary air pollution, focusing on emission reduction. In 2017, the European Commission warned BCC to clean the city’s air by 2020 or face fines. BCC received £1,130,982 from the Emergency Active Travel Fund in May 2020 with ~£500,000+ of match-funding to implement active travel measures, including LTNs, in partnership with Transport for West Midlands, under the “Places for People” initiative [45,46]. During the COVID-19 Pandemic, to expedite implementation, BCC used Emergency Traffic Regulation Orders, allowing for concurrent consultations and adjustments based on future public feedback [47]. Kings Heath, a Birmingham suburb favourably known for its commuter links, schools, and residential appeal, was selected for the LTN initiative due to concerns about pedestrian and cyclist safety, poor air quality, finite road space, and ‘rat running’ issues [48]. Community support for traffic-calming measures grew after the 2018 BBC documentary “Fighting for Air,” which highlighted air quality concerns in Kings Heath. Phase 1, completed in 2020, focused on West Kings Heath, with plans for Phase 2 entailing four packages of enhancements and new measures across the area by Autumn 2024 [46].

### 1.4. Aim of the Study

LTNs offer significant potential for mitigating air pollution and fostering sustainable urban development. Despite growing interest in LTNs in the UK, there is still a gap in knowledge regarding their effectiveness and impact on the local community, especially in localised contexts outside London, given current research has been predominantly London-centric [32].

While LTNs aim to reduce through-traffic and promote sustainable transport, they may inadvertently increase congestion and emissions in other areas [30,49]. Understanding public perspectives on LTNs is crucial for their successful implementation, ensuring they meet community needs which are understudied. Poor air quality directly affects public health and well-being, making improving air quality and promoting active transportation essential. By examining air pollution and public perspectives, research can determine if LTNs are suitable for adoption in other cities. Understanding the significance of LTNs could pioneer future innovative urban planning solutions in the UK.

Despite evaluations of LTN benefits for active transportation and equity, Aldred et al. [30] noted that their effects on air pollution and public perspectives remain understudied. The timing of the research may have coincided with the impacts of COVID-19, leading to a reduction in motor vehicle usage. Xiao et al. [34] noted that while evidence supports LTNs as effective in reducing driving and decreasing air pollution, most studies are supported within the ‘grey literature’ (in particular within council reports) and lack comparison groups, making it uncertain whether changes in transport behaviour were solely due to LTNs. Buehler and Pucher [50] stress the significance of community involvement and broad stakeholder participation in decision-making regarding initiatives to encourage cycling and walking, both during COVID-19 and over the long term. Xiao et al. [34] argued that past analyses struggled to differentiate the impacts of LTNs from concurrent major infrastructural changes. Aldred and Goodman’s [51] research on ‘mini-Holland’ in Waltham Forest illustrated this, where LTNs combined modal filters with enhancements like cycle paths and greening. However, this observation aligns with studies like Sabelis [52], which emphasise the need for multiple initiatives to ensure the success of LTNs.

To address these gaps, this paper uses the case study of Kings Heath LTN in Birmingham, UK. The city of Birmingham is currently undergoing a set net zero and transport decarbonisation plan aimed at tackling air pollution and encouraging active travel (see Section 2.1), with LTNs identified as potential interventions to contribute in this sense. The paper aims to evaluate the impact of Kings Heath LTN in reducing air pollution, through analysis of air quality data (NO_2_) following the introduction of LTNs, in conjunction with local community perspectives regarding their overall satisfaction of the scheme and perceived impacts on health and well-being. We hypothesise that the implementation of the LTN scheme in Kings Heath has led to a reduction in air pollution levels and an increase in active transportation but also faces challenges due to community acceptance and safety perceptions.

## 2. Materials and Methods

This study employs a convergent parallel mixed-method approach using quantitative and qualitative data to investigate the impacts of the LTN on Kings Heath and the community’s perspective on the scheme. Data collection methods employed for this study comprise air-quality-monitoring sensors (quantitative) and an online survey questionnaire (quantitative and qualitative). The convergent parallel design approach involves collecting quantitative and qualitative data at the same time, analysing them independently, and then exploring the results to determine whether the findings support or contradict each other [53]. The convergent parallel design approach was employed to provide a more comprehensive understanding of the impacts of the LTN, collecting richer data from both numerical data and contextual participants’ narratives related to opinions, perceptions, and potential resistances about the scheme. This approach allowed for triangulation between qualitative and quantitative data, with consequent enhanced analytical interpretation.

### 2.1. Study Location

Birmingham is a large metropolitan city located in the West Midlands region of the United Kingdom (UK). It is the UK’s second-largest city after London in terms of population, with a population of approximately 1.14 million and a population density of around 4100 people/km^2^ [54]. Birmingham is a young and diverse city, with more than 1 out of 5 residents aged 0–15, and almost 1 out of 3 residents are members of an ethnic minority [55]. Birmingham is also a major national economic hub, with an estimated contribution of £51.7bn in gross value added to the national economy in 2021 (3.1%) [55] and a GDP of £35.4bn in 2022. The income per capita in Birmingham is approximately £30,500, which is slightly below the national average [56]. Nonetheless, the city faces social and economic challenges, with pockets of deprivation and inequality making Birmingham the seventh most deprived local authority in England since 2015 [55].

Birmingham has an extensive transportation network, including buses, trains, a recently developed tram system that connects key areas within the city and the surrounding region, the Birmingham International Airport and a future rail hub for the HS2 high-speed railway [57]. Data from the National Travel Survey 2023 show that more than two-thirds of journeys are carried out by cars and vans, with walking accounting for around 23% and public transport 2.5%. Both active travel and public transport trends in Birmingham are below the national average and have shown constant declining rates over the last years [55]. In this context, Birmingham City Council has developed the Birmingham Transport Plan 2031 [57], with the commitment to reduce the negative impacts of transport on the environment and contribute to the city (and regional) net zero commitment. The plan aligns with the existing Clean Air Strategy [58] and the Walking and Cycling Strategy and Infrastructure Plan [59], and considers reallocating road space and prioritising active travel in local neighbourhoods as key principles in setting the plan.

### 2.2. Air Quality Monitoring

Air quality data in the form of NO_2_ was provided by BCC across locations within and surrounding Kings Heath LTN. Utilising diffusion tubes, known for their cost-effectiveness and ease of use, local authorities can monitor air quality in urban areas [60]. These tubes feature a permeable membrane to facilitate air diffusion, with molecules of NO_2_ reacting with an internal chemical reagent to form a measurable compound [61]. Concentration is determined by the degree of chemical reaction [62]. Months with skewed data were excluded from analysis and marked as “n/a” in the raw data.

The air quality data comprises monthly averages of NO_2_ (µg/m^3^) concentrations measured by passive diffusion tubes at 20 sites (Figure 1) between December 2020 to July 2023. An additional 5 sites were included with measurements spanning from August 2021 to July 2023. Data from 2 locations, BHM04 and BHM03, encompassed NO_2_ levels north of Kings Heath High Street for the years 2016, 2019, 2020, 2021, and 2022. Initial monitoring for LTN installation preparation was not provided by the BCC, except for BHM04 and BHM03. In adherence with DEFRA [63] guidance, NO_2_ monitoring sites were categorised as urban background if they were situated away from significant pollution sources, thus considered representative of pollutant concentrations in urban residential areas. Sites were classified as roadside if they were within 5 m of a main road’s kerb [31].

Mean annual NO_2_ levels, seasonal variations, projecting concentrations for North High Street, and evaluating air quality on LTN and non-LTN roads were examined. This method provided comprehensive air quality monitoring, crucial for assessing interventions like LTNs’ impact on urban NO_2_ levels while accounting for biases and seasonal anomalies [31]. Statistical testing was omitted from the analysis of air quality data due to its inability to account for seasonal variations, COVID-19 discrepancies, and the extensive volume of data.

### 2.3. Survey Questionnaire

#### 2.3.1. Procedure and Study Sample

Data for this study were collected through a survey questionnaire carried out in Birmingham between November and December 2023 to explore Kings Heath residents’ views regarding the implementation of the LTN scheme. To ensure the validity and reliability of the survey, a pilot survey was conducted with a small sample of diverse test subjects to identify and address any ambiguities or misunderstandings in the survey questions and gather feedback [64]. The received feedback was used to refine the questionnaire, including rewording some questions for better readability and removing or modifying misleading questions to maintain integrity and reduce bias [65].

The survey was shared with Kings Heath residents aged above 18 years old through word of mouth and link sharing in digital media outlets, including (1) “B13 Moseley, B14 Kings Heath and B12 Balsall Heath” Facebook group (~30,000 members); (2) “Kings Heathens Uncensored” private Facebook group (~3700 members); (3) “Kings Heath LTN concerns” public Facebook Group (~3700 members); and (4) local road WhatsApp group for neighbours. Weekly reminders were implemented to enhance participation and response rates, varying between weekends and weekdays at different times, which ensured a broad range of respondents. The survey returned 212 participants’ responses, which were reduced to 210, due to the incomplete submission of the survey from two participants. Table 2 provides an overview of the sample socio-demographic characteristics.

The finalised survey version was designed to be completed in approximately 30 min, which makes it sufficiently long to deter multiple responses from the same person, whilst also preventing responder fatigue which may adversely affect the quality of data [66]. Similarly, anonymity regarding a number of participants’ socio-demographic characteristics, such as name, postcode, and household income, was employed to encourage more responses. Ethical approval was received from the University of Birmingham School of Geography, Earth and Environmental Sciences Undergraduate Research Ethics Committee, with the study assessed to not raise any significant concern requiring further review. Before the completion of the online survey, participant consent was obtained, and participants were informed of the voluntary nature of their involvement in the study, provided with a summary of the project and a participant information sheet, ensuring transparency and upholding ethical research practices throughout the data collection process.

#### 2.3.2. Measures

To enhance respondent engagement and minimise survey fatigue, the questionnaire included a mix of interactive open-ended statements and multiple-choice questions, along with graphics to engage respondents at the beginning of the survey [67]. The following Section outlines the parts of the questionnaire that were analysed in the present article.

Socio-demographic background information includes age, gender (male, female, non-binary, prefer not to say), relationship with Kings Heath (current resident, past resident, living in the neighbouring area, work, visit, no connections), and length of residence (in years), see Table 2.

Opinions about the LTN scheme comprise participants rating their opinion about the LTN in Kings Heath on a five-point rating scale (1 = “very positive”, 2 = “positive”, 3 = “neutral”, 4 = “negative”, 5 = “very negative”). Participants were also asked to provide a percentage reduction in air pollution that would be significant enough to influence their opinion and acceptance of LTNs (open-ended question). Moreover, they were provided with a list of features they would consider essential for an ideal LTN (pedestrian-friendly; cyclist-friendly; efficient public transport; adequate parking facilities; green spaces and environmental considerations; traffic management measures; community engagement and consultation; consideration for emergency service access; air quality and pollution control; health and well-being considerations; nothing would make LTNs ideal).

Perceived health impacts of the LTN include potential positive or negative impacts on respondents’ health (open-ended question) and impacts on the local air quality, measured on a five-point rating scale (1 = “significant positive”, 2 = “positive”, 3 = “no noticeable”, 4 = “negative”, 5 = “significant negative”).

Travel behaviour and barriers include changes in the main travel mode used pre- and post-implementation of the scheme (for walking, cycling, car, public transport, and taxi) and factors influencing potential modifications (open-ended question). Participants were also asked about potential changes in traffic volume due to the LTN (1 = “significant increase”, 2 = “not significant change”, 3 = “reduction”, 4 = “not sure”), increased journey length due to the LTN (yes, no, not sure), and individual safety as a pedestrian/cyclist (1 = “much safer”, 2 = “somewhat safer”, 3 = “no change”, 4 = “less safe”, 5 = “much less safe”).

Finally, respondents were asked to provide any additional information regarding issues experienced because of the LTN implementation and potential suggestions or modifications to the scheme (open-ended question).

#### 2.3.3. Data Analysis

Data collected from the survey were analysed using the software IBM SPSS Statistics 27. The analysis comprised descriptive statistics, including frequency and cross-tabulations. A Pearson’s chi-squared test and Kruskal–Wallis *t*-test for independent samples were used as appropriate to test group differences associated with individual opinion regarding the LTN and selected variables including age, gender, perceived health impacts of LTNs, air quality, increased journeys, increased traffic volume, and safety.

For the open-ended questions, qualitative thematic coding and analysis were carried out using the software NVivo 12, using the procedure outlined in Robson and McCartan [68]. The process included collecting participants’ statements and carefully reading to identify initial ideas. An initial set of categories was identified by grouping and coding extracts of text with similar content, which were further refined into the final set of themes and sub-themes.

## 3. Results

### 3.1. Air Quality Monitoring

Figure 2 shows that South High Street (31.02 μg/m^3^) and Springfield Road (29.99 μg/m^3^) had the greatest mean NO_2_ concentrations over the 3-year period. York Road and Valentine Road saw higher concentrations in 2022 compared to 2021. Wheelers Lane (12.84 μg/m^3^) and Tenbury Road (12.86 μg/m^3^) recorded the lowest annual mean concentrations in 2023. Avenue Road witnessed the most significant reduction, decreasing by 17.7%. NO_2_ levels peaked above 40 µg/m^3^ on Springfield Road in January 2022 (42.21 μg/m^3^) and South High Street in March 2022 (40.69 μg/m^3^).

From 2020 to 2023, monthly NO_2_ concentrations consistently decreased in summer compared to winter, meeting the 2005 WHO AQG but not the 2021 AGQ. Summer 2021 levels were 16.15 µg/m^3^, 28.7% lower than the 2020–2021 winter average of 22.39 µg/m^3^. Reductions continued in subsequent years: 25.3% in 2021–2022 and 29.5% in 2022–2023 (Figure 3).

Mean NO_2_ concentrations on North High Street declined from 40 to 27.65 µg/m^3^ between 2016 and 2022, with a notable drop to 27.1 µg/m^3^ in 2020. The average annual decrease rate from 2016 to 2022 is approximately 5.2%. If the trends persist in a linear fashion, NO_2_ concentrations would be expected to reach 10 µg/m^3^ by 2028 (Figure 4). However, it is noted that a linear forecast is a simple model without physical basis, and its extrapolation will have high uncertainties.

In December 2020, LTN roads averaged 23.65 µg/m^3^, lower than non-LTN roads (25.01 µg/m^3^, +5.8%). By July 2023, LTN roads averaged 9.52 µg/m^3^, compared to non-LTN roads (13.13 µg/m^3^). February 2022 showed declines on both LTN and non-LTN roads to 14.46 µg/m^3^ and 13.01 µg/m^3^. LTN Roads had an average of 17.39 µg/m^3^ across the period, while non-LTN Roads averaged 20.86 µg/m^3^ (Figure 5).

### 3.2. Survey Questionnaire

#### 3.2.1. Opinions Regarding the LTN Scheme

Table 3 shows that around a third of the respondents (31.5%) reported having positive opinions about the implementation of the Kings Health LTN scheme, while more than half (60.5%) reported a negative opinion, with 35.7% having a very negative opinion. Gender differentiation shows that, overall, men have a more positive opinion compared to women. In terms of age differentiation, older respondents (65+) generally expressed more negative views (65.9%) than positive (29.6%). Across all age ranges, the median response was negative, except the 35–44 range with a median neutral opinion. Younger respondents (18–24) and the 65+ age groups generally expressed more negative than positive opinions, while the 35–44 and 45–55 age groups exhibit the highest percentage of positive opinions.

Participants were also asked what criteria would be considered essential for an ideal LTN in Kings Heath. Figure 6 shows that the top-ranked criteria include pedestrian friendliness, public transport, traffic management, and emergency services, with less emphasis on community engagement, health and well-being, and green spaces. Parking was the least prioritised. Moreover, almost a quarter of the respondents expressed that nothing would make LTNs ideal, stressing their negative opinion regarding the scheme.

Similarly, participants were asked what level of reduction in air pollution would make the LTN scheme acceptable. Table 4 shows that among 164 responses (out of 210), the average percentage of reduction is around 32.0%. Percentages differ significantly between respondents with different opinions regarding the LTN scheme. The majority of respondents with positive opinions expressed low percentages of reduction, predominantly between 0/0 and 20.0%, and overall were more inclined to accept LTNs regardless of the decrease in pollution. On the other hand, the percentage of acceptance increases with negative opinions regarding LTNs, with around 10.0% reporting a value between 80.0 and 100.0% reduction and 14.0% stating that any reduction in air pollution would make the scheme worthy. Other participants used this as an opportunity to voice their opinion that a reduction in pollution is insufficient unless it applies universally, rather than solely affecting those living on LTN-blocked roads: “No one would welcome air pollution but sending it all into other areas is not the solution” (Respondent 32).

#### 3.2.2. LTN’s Impacts on Health

Almost half of the participants (42.4%) stated that LTNs had negative impacts on their health, while 38.6% noted no changes and 9.0% positive impacts. Similar trends are found for the impacts of the LTN on air quality, with 44.7% reporting negative impacts, 30.0% no changes, and 25.3% positive ones. Data indicate a correlation between general opinion about the LTN scheme and these potential impacts on both health (*p* < 0.001 K-W H test) and air pollution (*p* < 0.001 K-W H test). Figure 7 shows that positive health impact correlated with positive views of the LTN scheme (90.0%), while, on the other hand, negative impacts correlated with negative views (91.0%), with also 53.1% reporting no changes among those with negative opinions. Similarly, Figure 8 shows similar trends regarding the impacts of the LTN on air pollution, especially regarding those reporting significant positive impacts (89.5%) and significant negative ones, with more than half (58.7%) reporting no changes among those with negative opinions.

Respondents expressed concerns about worsened physical health due to increased traffic and pollutant levels, exacerbating pre-existing conditions like asthma and respiratory issues: “Significant negative impact on my health (getting worst as I’m suffering chronic disease) as the road I’m living has been significantly polluted with heavy/re-routed traffic” (Respondent 118). Respondents noted changes in habits, correlating with increased pollutants accumulating on windows, especially for those living on main or boundary roads: “Negative [impact on health], because all the traffic is forced down our road with traffic bumper to bumper at certain times of the day. Therefore, we can’t open the windows” (Respondent 10). On the other hand, respondents with a positive opinion of the scheme reported improved sleep quality and breathing, along with increased physical exercise: “I live on a rat run street and have a child who before the LTN had repeated trips to the hospital because of asthma. There is now less car pollution” (Respondent 175).

Participants highlighted negative impacts on their mental health because of the LTN implementation, experiencing issues such as stress or anxiety, often linked to navigation challenges and feelings of isolation, discouraging them from leaving home: “I cannot avoid travelling through Kings Heath on my commute. The stress of navigating this route and the fact I have to get up for work a lot earlier to get through it, has significantly harmed my physical and mental wellbeing. The breakdown of a community which I have lived in all my life has also been damaging for my mental health” (Respondent 144). Respondents expressing positive opinions often cited improved mental well-being due to increased physical activity, reduced pollution, and enhanced safety. This was especially notable among residents on side roads affected by the scheme’s blockades: “[Yes Positive Impact on] mental health! It’s so much better cycling when I’m not having to be in constant high alert as to what drivers are going to do” (Respondent 72).

#### 3.2.3. Changes in Travel Behaviour and Transport Barriers

Looking at changes in how residents travelled before and after the implementation of the LTN scheme, Figure 9 shows a 11.9% decrease in car use and 7.5% in public transport, with an increase in walking and cycling trends of 11.0% and 6.2%, respectively. Negative transportation impacts resulted from increased traffic congestion, longer commute times, difficulties in navigating diverted routes, and reduced accessibility for certain transportation modes: “Buses have become less reliable… I used to travel by bus. Now I travel by car” (Respondent 154). On the other hand, participants valued enhanced cycling infrastructure, heightened safety within LTN zones, and the preference for walking over driving due to quicker travel times: “I drive considerably less and cycle considerably more” (Respondent 111).

Changes in traffic volume were found to be associated with increased traffic by two-thirds of the respondents (66.2%), with 18.1% mentioning no changes and only 3.8% mentioning decreasing trends. Common issues reported in this sense were linked to the changes in traffic flow, availability of parking and idling behaviours: “Worse traffic congestion because same traffic on fewer roads. More parking issues. Worse air quality because more standing traffic in congested roads. More ‘road rage’ and inconsiderate driving as motorists get frustrated in traffic jams caused by LTN” (Respondent 70). A correlation between opinions on the LTN and changes in traffic volume (Figure 10) (*p* < 0.001 K-W H test) shows that increased traffic was reported predominantly by those with negative opinions (83.5%), while reduction or no changes by those with positive opinions (100% and 79.0%, respectively).

Linked to changes in traffic volumes, two-thirds of the respondents reported also experiencing longer journeys, with increased awareness of air quality depletion: “Certain road closures to ‘benefit air pollution’ has caused the traffic to be busier and with other cars having to wait longer in the traffic it could potentially have a negative impact on the air pollution…” (Respondent 130); Figure 11 shows that a correlation between opinions on the LTN and longer journeys (*p* < 0.001 K-W H test) shows again that the vast majority of those experiencing longer journeys have negative (87.5%), while 87% of those not experiencing longer journeys have positive opinions.

Participants were also asked about their perceived road safety and linked with the LTN scheme. Around a third (33.8%) reported no changes in this sense, while 22.7% mentioned feeling much safer and 19.8% somewhat safer. Participants stressed the ability to cycle and walk safely within LTN zones: “Much more safer to cycle” (Respondent 98). On the other hand, around a third of respondents mentioned feeling less safe. Main issues experienced in this sense are linked with car drivers’ behaviours: “I think that due to car drivers getting stressed over lack of traffic flow around Kings Heath, it has indirectly caused them to drive more erratically and increased dangers on the main road. (Jumping traffic lights, bad parking, point turns in inappropriate places) which in turn endangers pedestrians and cyclists more” (Respondent 43). Again, a correlation between opinions on the LTN and perceived safety (*p* < 0.001 K-W H test) shows that 100.0% feeling much less safe have a negative opinion of the scheme, while on the opposite side, 87.3% feeling much safer have positive opinions (Figure 12).

#### 3.2.4. Community Issues Experienced Due to LTN Scheme Implementation

In addition to transport and mobility issues linked with the LTN scheme, participants were asked to provide further information regarding additional issues. A strong theme that emerged was the division that the implementation of the LTN has created within the community of Kings Heath. Several respondents reported increased disharmony between community members, including on social media, disrupting what was considered a positive community: “I feel that the way it was implemented really divided the community and created ill feeling between those who benefit from reduced traffic and those of us who gained the traffic which travels through the area on our roads.” (Respondent 15). The main division has been experienced between those in support of and against the scheme, and those living within/outside the scheme boundaries.

Some respondents in this sense reported a loss of attractiveness of the area, with fewer friends visiting their houses, and a perceived decrease in house price because of this issue: “I pay more council tax for my house in the area which also happens to be on one of the main roads, which has not had the benefit of becoming a ‘blocked off road’. Instead, we have become a much busier road, more pollution, angry drivers, constant traffic, practically hell around peak school hours for pick up, decreasing the value of our properties. This has been shown by speaking to neighbours who have tried to sell houses with no luck even though the location itself was otherwise amazing. We have lost neighbours who have lived here decades”. (Respondent 2)

Moreover, respondents reported a surge in anti-social behaviour, vandalism to LTN planters and bollards, and an intensified drug trade, contributing to a prevailing sense of insecurity: “Creates anti-social behaviour as roads are closed to traffic and increases vandalism. Feels unsafe to go out when it’s dark along the closed roads as lighting is still very poor and no one is about” (Respondent 166).

Some of the respondents, including people involved with local business, observed also a downturn in business activity, leading to a transformation of the retail landscape in Kings Heath and a less appealing high street as a consequence: “Think it’s had a detrimental effect to businesses in Kings Heath and the appeal of shopping on the High Street generally. The LTN roads are quieter but the traffic has been diverted into another place which has caused mayhem” (Respondent 12).

#### 3.2.5. Modifications to the LTN Scheme

Participants were finally asked to provide suggestions for improvements or modifications to the current LTN setup. While a significant number of respondents among those with negative opinions suggested removing the current LTN system altogether, some participants sought baseline pollution levels before the implementation: “Get rid. Take baseline pollution levels without LTNs implementation of traffic management plan including new rail stations. Gradually reintroduce LTNs with pollution/traffic monitoring” (Respondent 8). Moreover, participants mentioned potential modification following the opening of the train station in Kings Heath and Stirchley (which were supposed to be opened for the 2022 Commonwealth Games) and further expansion of the cycle lanes network.

Most respondents suggested traffic-calming methods, including one-way streets, better management of traffic lights in relation to changes in traffic, and speed bumps. Moreover, respondents suggested enforcing parking regulations or improving parking on the high- street, which coincided with an improvement in public transport, more specifically bus routes: “Enforce parking regulations in spaces next to bollards—these are often used by cars, causing obstruction for cyclists and pedestrians” (Respondent 19). In this sense, the issue of parking and idling around local schools emerged strongly: “There are a number of schools in the Kings Heath area, particularly primaries. The working pattern of parents means that many need to use a car to drop off children, sometimes at different locations, before driving to work. These parents can’t simply switch to buses as the planners of the LTN scheme would like. Buses can only accommodate two pushchairs and so are difficult for parents with young children to use. An improvement could be to have a shuttle school bus service to and from schools with a driver and a guide on each bus” (Respondent 191).

Those reporting vandalism as a key issue proposed fixed bollards as a solution, especially to make York Road fully pedestrianised: “York Road is now buzzing on a sunny summer evening, thanks, almost exclusively to the LTN. That’s what makes it so great!” (Respondent 53). Other respondents proposed instead foldable bollards to ensure traffic flow during peak times or emergencies. Finally, respondents requested modifications to urban design and urban furniture to make the area more attractive for active transport: “I think people might appreciate them more if they were accompanied by urban realm improvements such as attractive paving and planting. It would also be great to ban parking on the high street to reduce traffic there, or ideally pedestrianise it to make spending time there even more pleasant” (Respondent 40).

## 4. Discussion

This paper explored the impacts of LTNs in reducing air pollution and local community perspectives regarding their overall satisfaction with the scheme and perceived impacts on health and well-being in Kings Health, Birmingham. This study provides additional insights into the limited research on LTNs outside the context of London. Moreover, it expands the work of Pritchett et al. [32] regarding Kings Heath LTN, by providing an overview of air-quality-monitoring data, integrated with Kings Heath residents’ opinions over a different timeline (i.e., two years following BCC consultations’ pre- and post-LTN implementation). The findings from both air quality monitoring and the survey confirm our hypothesis that the LTN scheme has led to a reduction in air pollution levels and increased active travel but also created issues within the Kings Heath community related to the overall scheme acceptance.

The analysis of air quality data reveals a reduction in NO_2_ levels within Kings Heath in the years after the implementation of LTNs. The decrease in NO_2_ concentrations suggests a positive impact on air quality, potentially attributed to reduced vehicular traffic and emissions within the designated LTN zones. The measured NO_2_ concentrations were all less than the 40 μg/m^3^ required by the 2010 UK AQSs. However, when compared to the 2021 WHO AQGs of 10 μg/m^3^ for NO_2_, air quality rarely meets the recommended levels. Many respondents believed the area was already compliant with government guidelines, hence diminishing the perceived need for improvement despite inconveniences attributed to the LTNs. Projected decreases in annual concentrations suggest a potential alignment with the 2021 WHO AQG by 2028, which assumes continuing linear reductions, stressing the importance of future initiatives. It is worth noticing that NO_2_ data also do not take into account factors such as the COVID-19 pandemic, remote-work trends, broader transportation, the clean air zone implemented in 2021, and seasonal variations, which may have influenced air quality levels during the study period. Winter months would also see lower concentrations due to stagnant air masses, increased heating emissions, and reduced sunlight, as mentioned by Bodor et al. [69]. The absence of comprehensive baseline data makes it challenging to attribute the observed changes solely to the LTN scheme. Yang et al. [31] highlighted the importance of pre-LTN data for successful implementation. To accurately assess the impacts of future LTN schemes or modifications, it is crucial to establish comprehensive baseline data on air quality, traffic patterns, and community perspectives before implementation. This will enable more robust comparisons and attribution of observed changes.

The LTN has created a polarisation effect linked with the divergence in public opinion regarding the scheme. The findings from the study show a bimodal distribution of sentiments regarding the LTN, with the community broadly split between those with positive or negative sentiments and a limited number of those having neutral views. In this sense, residents’ experiences have influenced positively or negatively their opinions and acceptance of the scheme. A clear example in this sense is provided in Table 4, in which residents with a positive outlook on the scheme have more realistic expectations and associated acceptance of the extent to which the LTN can reduce air pollution. In contrast, those with negative views mentioned unrealistically large decreases in pollution levels. Such polarisation has led to a divisive effect on the community. Several participants highlighted how the topic of the LTN has generated acrimony, creating a depressing strain on once-positive community relations. On the one hand, those residents living within the LTN boundary reported positive opinions of the scheme, mentioning positive physical and mental health, improvements in air quality levels, increased walking and cycling, and personal safety trends. On the other hand, residents living outside the modal filters expressed feelings of discrimination, reporting a deterioration of air quality, linked mainly to increased volume of road traffic and idling, stressing the injustice of shifting pollution from one location to another. The increase in traffic was also associated with negative mental health, due to the challenges of navigating in Kings Heath and longer journeys, with also safety issues reported due to poor driving behaviour of car users. Many respondents expressed frustration over lack of consultation and poor implementation, feeling ignored by authorities. The significant proportion of negative sentiment, coupled with the frustration expressed by respondents over the perceived lack of consultation and disregard for their concerns, underscores the need for more inclusive and transparent decision-making processes. The role of public participation in the success of LTNs was highlighted by different studies [31,70], and to enhance success, government bodies must prioritise listening to public consultation. King Heath’s response to LTNs reveals underlying tensions and unresolved issues, emphasising the imperative for effective communication, genuine consultation, and collaborative decision-making to navigate complex urban challenges.

The shift in transportation behaviour following the implementation of LTNs, particularly the decrease in public transportation usage attributed to unreliable bus services and congestion, reflects the “carrot” and “stick” effects discussed by Laverty et al. [33] and Xiao et al. [34]. This has led to increased walking and cycling, aligning with the intended goal of promoting active transportation and highlighting LTNs as effective in incentivizing alternative modes of transport, potentially leading to long-term health and environmental benefits. However, concerns about safety have led to a shift from walking to driving for some residents, especially outside LTN boundaries, suggesting potential unintended negative consequences for the population. This aspect underscores the importance of considering diverse perspectives of the population and potential drawbacks when evaluating the overall impacts of the LTNs. Contrary to other studies, the LTN was implemented without the integration with complementary initiatives related to public transport (e.g., the opening of Kings Heath train station, increased bus routes, and subsidised fares) and cycling infrastructure (e.g., cycle hire schemes). Participants highlighted this issue, especially about public transport, and provided a number of measures that could improve LTN acceptance among the community. Emphasis was put on stricter parking regulations, traffic-calming measures, partial road closures, and one-way systems, allowing traffic to filter through and reduce poor drivers’ behaviour, Similarly, fixed or flexible bollards were suggested to reduce vandalism acts, together with improvements in urban design to incentivise active travel.

This study has a few limitations. Limited baseline data made it challenging to assess spatial pollutant variability, especially regarding potential LTN impacts on boundary roads. This hindered comparisons, both with our data and previous research, considering the proximity to emission sources and the determination of external factors influencing our data. In future studies, it is crucial to explore differences between residents impacted by LTNs and those outside the scheme shaping public perceptions and health impacts. Studying seasonal variations and their influence on transport use could shed light on whether decreased car usage significantly affects NO_2_ emissions. The survey response rate and distribution may not accurately represent the entire Kings Heath community, potentially introducing sampling biases. Consequently, the analysis of the survey may appear biassed towards negative feedback simply because a higher proportion of respondents provided negative responses to open-ended questions. Moreover, the survey was distributed through online channels, with potential implications of digital exclusion in the sample. Findings on perceived health impacts rely on self-reported data, which may be influenced by individual biases and perceptions, suggesting a need for also integrating more objective health metrics or longitudinal studies for more robust evidence of the LTN impacts. Finally, the impact of the COVID pandemic on transportation lifestyles, including increased remote work, raises doubts about the true representation of air quality data. Consequently, conducting in-depth statistical analyses of air quality data became challenging, as changes over time were not adequately accounted for. Thus, excluding data from the pandemic years may be necessary for more accurate assessments.

## 5. Conclusions

Air pollution persists as a longstanding environmental challenge, with LTNs representing a potential urban initiative to improve air quality and promote active travel. Existing research by Nieuwenhuijsen [42] emphasises the importance of thoroughly evaluating LTN schemes for their effectiveness, acceptability, and impacts on health, liveability, and sustainability. With limited comparable studies on this relatively new and understudied initiative, it is essential to understand its impact on air quality and the local community, including perceptions of health, safety, and behavioural changes, as noted by Aldred et al. [30]. As initiatives like Places for People are implemented in Kings Heath, it becomes crucial to assess both the successes and challenges faced by LTNs. The significance of public perceptions in initiatives to encourage cycling and walking rather than vehicular transport was highlighted by Buehler and Pucher [50].

Our study aligns with Sabelis [52], who highlights the importance of complementary measures alongside the LTNs, which leads us to accept our hypothesis. Analysis of air quality data revealed a reduction aligning with 2005 standards but not complying with the 2021 guidance of 10 μg/m^3^, and a potential alignment with the 2021 AQG was projected by 2028, indicating the need for additional initiatives alongside LTNs to promote electric or hybrid vehicles and improve walking and public transportation spaces. Seasonal variations in air quality may suggest reduced vehicular traffic during warmer months, although external factors could also play a role. The lack of pre-LTN data complicates the attribution of changes, as emphasised by Yang et al. [31], highlighting the importance of baseline data for successful implementation. The negative response rate of residents in our study was relevant, comprising two-thirds of respondents. This sentiment, combined with concerns about boundary road congestion, safety, pollution, accessibility, and distrust in local authorities, highlights the polarising and divisive impact of LTNs, along with the unintended consequences of the scheme.

Overall, this study provides valuable insights for local urban planners and policymakers, highlighting the potential of LTNs to tackle air pollution and increase active travel, with associated health and wellbeing impacts. The study uses a mixed-method approach comprising quantitative and qualitative data collected from air-quality-monitoring sensors and a survey questionnaire. Such a holistic approach allows for a more nuanced understanding of the LTN impacts, with qualitative data providing residents’ perceptions and experiences, revealing the social and psychological impacts of LTNs. The focus of this study on a context outside London, which currently comprises the bulk of the existing knowledge on the topic, expanding the emerging work on LTN impacts. It confirms some of the findings from Pritchett et al. [32] whilst also providing new information about air quality reduction in Kings Heath and the community perspective over a longer timeline since the implementation of the LTN scheme.

The LTN in Kings Heath has led to increasing walking and cycling trends and received positive feedback on roads like York Road, with overwhelming appraisal and suggestions for it to be even more pedestrianised. At the same time, the findings from the research highlight that LTNs can be controversial and have the potential to disrupt communities due to traffic redistribution, accessibility, perceived economic impacts, and inequalities. These issues stress the need for improved public consultations, especially with those who might be negatively impacted by the scheme, to build consensus and enhance the success of the LTN.

More research on LTNs is needed outside the London area to explore how these low-emission zones influence travel behaviour and encourage a modal shift towards more sustainable transport options, particularly active travel. Moreover, further investigation is needed to understand the effectiveness of LTNs in reducing traffic congestion versus displacing traffic to other areas, given the controversial outcomes mentioned in this study. Longitudinal studies evaluating the physical, mental health, and wellbeing benefits of active travel would provide valuable insights into Kings Heath’s development over the long term. Similarly, research should investigate the health impacts of potential reductions in car accidents. From an environmental perspective, future research on LTNs should examine long-term air quality changes and their integration into urban climate change and net zero plans at local, regional, and national scales. For Birmingham specifically, for example, given the lack of integration with broader initiatives crucial for success, future studies should examine baseline data before the planned opening in 2025 of Kings Heath, Moseley and Stirchley train stations [71], amongst other urban planning initiatives. From a social perspective, this study confirmed the controversial and divisive nature of LTNs. Future research should conduct longitudinal evaluations of perceptions, attitudes, and resistance towards LTNs over time, identifying key factors affecting these changes and community identity. Moreover, research is needed to determine if LTNs effectively reduce transport and social inequalities or if they exacerbate these issues due to poor consultation mechanisms and divisive community impacts.

## Figures and Tables

**Figure 1 ijerph-21-01638-f001:**
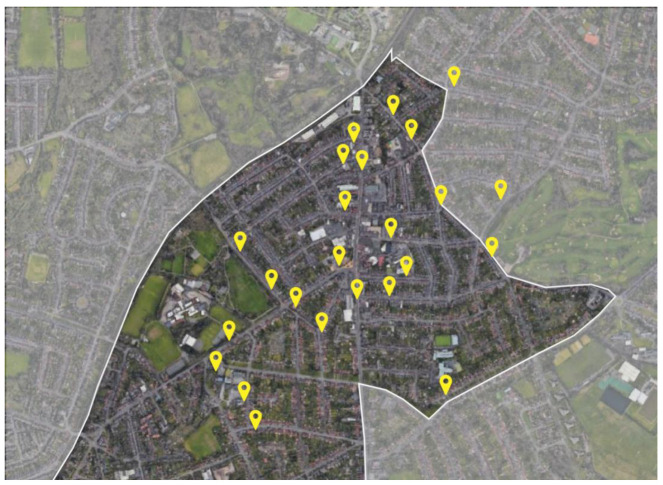
Air quality sensor passive diffusion tubes’ location in Kings Heath.

**Figure 2 ijerph-21-01638-f002:**
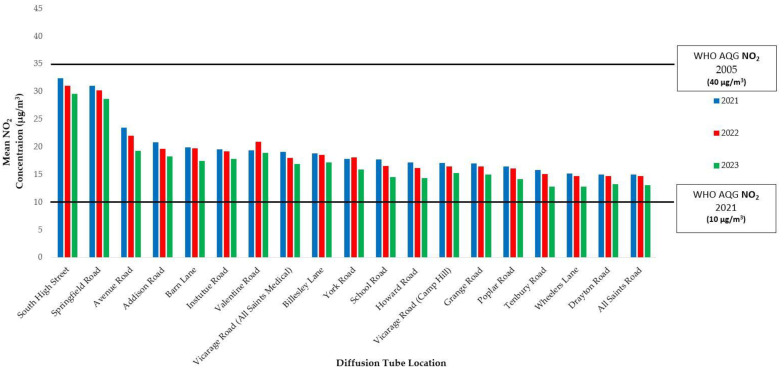
Mean annual nitrogen dioxide (NO_2_) concentrations (μg/m^3^) across 19 locations in Kings Heath for January 2021–July 2023, referencing the WHO AQG of 2005 (40 µg/m^3^) and 2021 (10 µg/m^3^). It is noted that the 2010 UK AQS is the same as the WHO AQG of 2005.

**Figure 3 ijerph-21-01638-f003:**
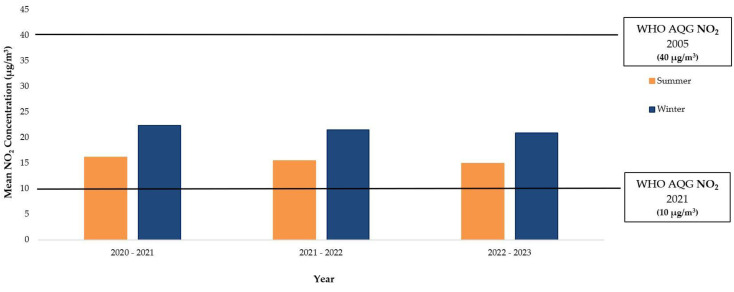
Mean monthly seasonal nitrogen dioxide (NO_2_) concentrations (µg/m^3^) comparing winter and summer months from 2020 to 2023, referencing the WHO AQG of 2005 (40 µg/m^3^) and 2021 (10 µg/m^3^). It is noted that the 2010 UK AQS is the same as the WHO AQG of 2005.

**Figure 4 ijerph-21-01638-f004:**
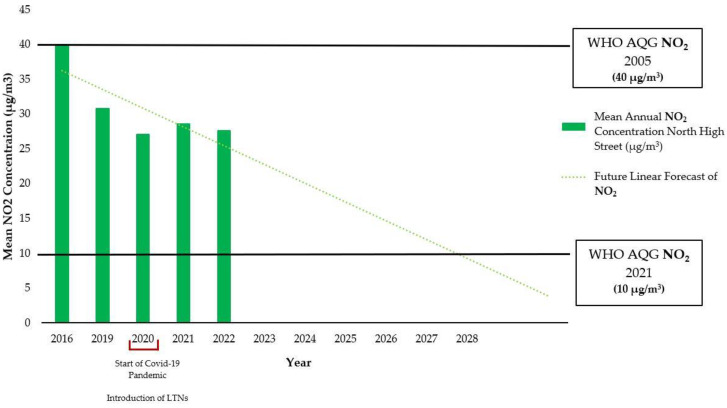
Projected mean annual NO_2_ concentrations (µg/m^3^) on North High Street, Kings Heath, from 2016, 2019, 2020, 2021, and 2022. The future linear forecast is represented by a green dotted line. In 2020, Kings Heath introduced Phase 1 LTNs alongside the onset of the COVID- 19 pandemic. Referencing the WHO AQG of 2005 (40 µg/m^3^) and 2021 (10 µg/m^3^). It is noted that the 2010 UK AQS is the same as the WHO AQG of 2005. Data from Sites 26 and 27.

**Figure 5 ijerph-21-01638-f005:**
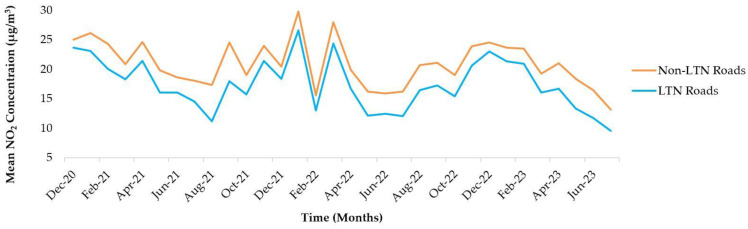
Monthly mean nitrogen dioxide (NO_2_) concentrations (µg/m^3^) on LTN and non-LTN Roads in Kings Heath from December 2020 to July 2023.

**Figure 6 ijerph-21-01638-f006:**
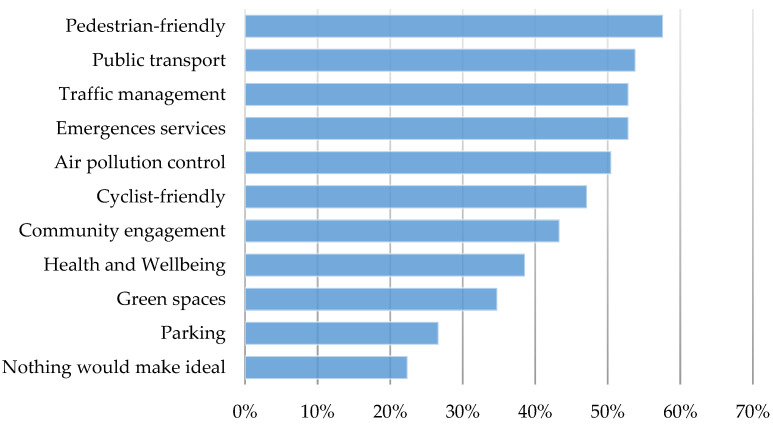
Essential criteria for LTN acceptance.

**Figure 7 ijerph-21-01638-f007:**
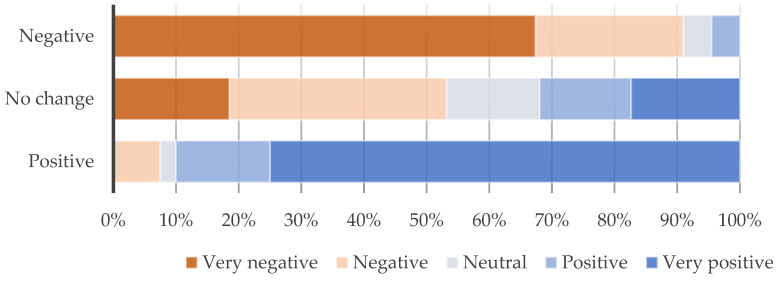
LTN impacts on individuals’ health by general opinion on LTN; *p* < 0.001 (Kruskal–Wallis H test).

**Figure 8 ijerph-21-01638-f008:**
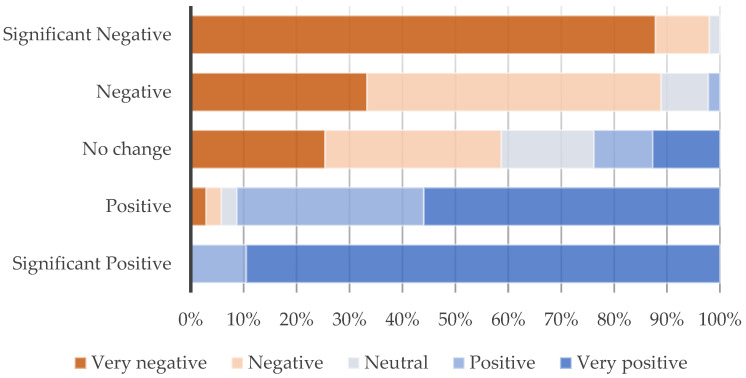
LTNs’ impacts on the local air quality by general opinion on LTN; *p* < 0.001 (Kruskal–Wallis H test).

**Figure 9 ijerph-21-01638-f009:**
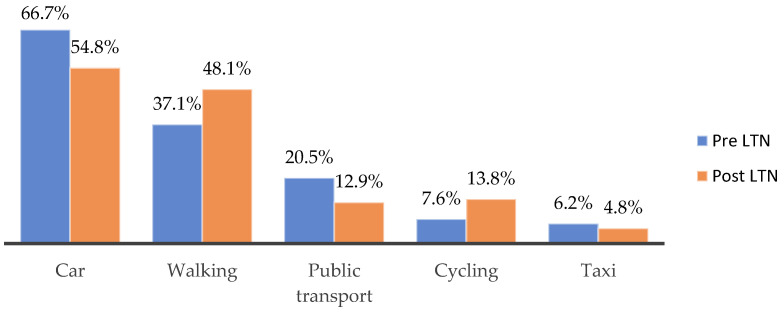
Primary mode of transportation into Kings Heath before and after the implementation of LTNs.

**Figure 10 ijerph-21-01638-f010:**
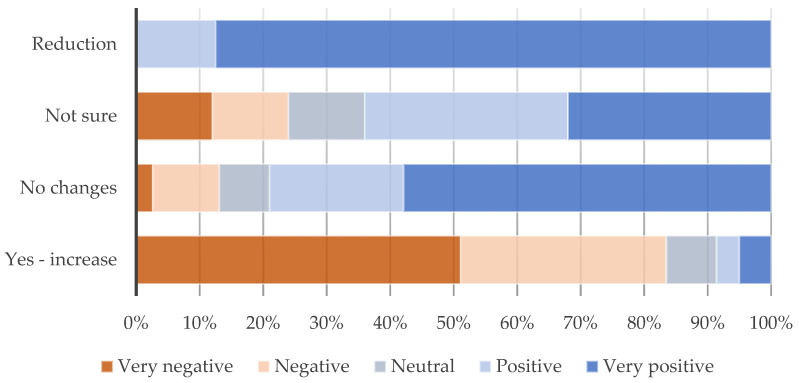
Changes in traffic volume due to LTN by general opinion about LTN; *p* < 0.001 (Kruskal–Wallis H test).

**Figure 11 ijerph-21-01638-f011:**
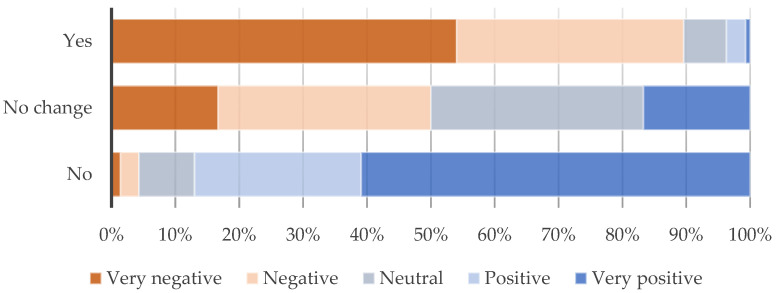
Experiencing longer journeys into Kings Heath due to the LTN by general opinion on LTN; *p* < 0.001 (Kruskal–Wallis H test).

**Figure 12 ijerph-21-01638-f012:**
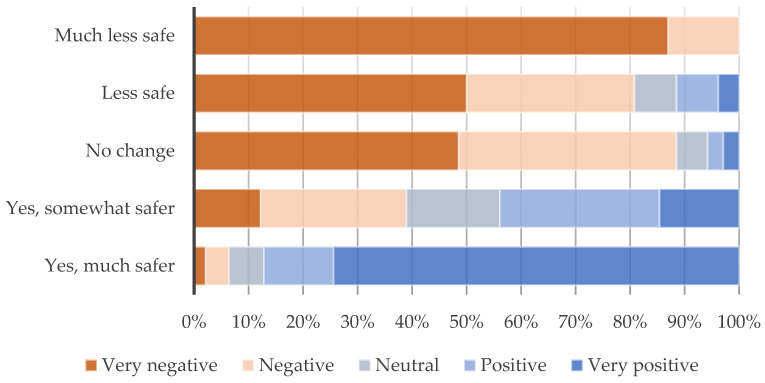
Safety as a pedestrian/cyclist within the LTN areas by general opinion on LTN; *p* < 0.001 (Kruskal–Wallis H test).

**Table 1 ijerph-21-01638-t001:** Comparison of pollutant levels from the air quality guidelines from WHO and UK Government (adapted from [10]).

Pollutant	Averaging Time	2005 WHO AQGs	2021 WHO AQGs	2010 UK AQSs
PM_2.5,_ µg/m^3^	Annual	10	5	20
	24 h	25	15	-
PM_10_, µg/m^3^	Annual	20	15	40
	24 h	50	45	50
O_3_, µg/m^3^	Peak season	-	60	-
	8 h	100	100	100
NO_2_, µg/m^3^	Annual	40	10	40
	24 h	-	25	200
SO_2_, µg/m^3^	24 h	20	40	125
CO_2_, mg/m^3^	24 h	-	4	-

**Table 2 ijerph-21-01638-t002:** Sample socio-demographic characteristics.

Socio-Demographic Characteristic	Percentage
Gender	
Male	42.9
Female	53.8
Prefer not to say	1.4
Non-binary	1.9
Age	
18–24	12.4
25–34	9.5
35–44	14.3
45–54	21.4
55–64	21.4
65+	21.0
Connection with Kings Heath	
Current Resident	63.8
Past resident lived in KH	5.2
Live in neighbouring area	28.0
Work	1.0
Visit	1.0
No Connection	1.0
Length of residence	
Less than 1 year	2.2
1–5 years	8.7
6–10 years	8.7
More than 10 years	80.4

**Table 3 ijerph-21-01638-t003:** Overall opinion about LTN by gender and age.

	Very Negative	Negative	Neutral	Positive	Very Positive
General sample	35.7%	24.8%	8.1%	10.5%	21.0%
Gender ***					
Male	37.8%	13.3%	6.7%	12.2%	30.0%
Female	34.5%	34.5%	9.7%	8.8%	12.4%
Prefer not to say	0.0%	0.0%	0.0%	33.3%	66.7%
Non-binary	50.0%	25.0%	0.0%	0.0%	25.0%
Age groups					
18–24	23.1%	46.2%	15.4%	7.7%	7.7%
25–34	40.0%	20.0%	5.0%	10.0%	25.0%
35–44	16.7%	30.0%	10.0%	16.7%	26.7%
45–54	31.1%	26.7%	2.2%	11.1%	28.9%
55–64	44.4%	17.8%	13.3%	8.9%	15.6%
65+	50.0%	15.9%	4.5%	9.1%	20.5%

* *p* < 0.01 (Chi-squared test).

**Table 4 ijerph-21-01638-t004:** Percentage of air pollution reduction to accept LTN scheme.

Percentage of Reduction	Negative	Neutral	Positive	Total
0–20	9.4%	35.3%	65.2%	29.0%
20–40	11.0%	35.3%	4.5%	11.0%
40–60	21.3%	0.0%	9.1%	15.7%
60–80	4.7%	0.0%	1.5%	3.3%
80–100	10.2%	0.0%	0.0%	6.2%
None	14.2%	0.0%	0.0%	8.6%
N/A *	29.1%	29.4%	19.7%	26.2%
Average	48.74	27.50	11.26	32.37
St Dev	27.18	21.05	17.47	23.99

* N/A includes answers that did not provide a percentage and responses like “I don’t know”.

## Data Availability

Air quality data used in the study are provided by Birmingham City Council and can be accessed from the following links: Air quality monitoring in Kings Heath—update February 2023 and Places for People—Diffusion Tube Data July 2023. Anonymised data supporting the conclusions of this article will be made available by the authors on request.

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
