# Peer review of "Assessing the Polarising Impacts of Low-Traffic Neighbourhoods: A Community Perspective from Birmingham, UK"

_ijerph, 2024, doi:10.3390/ijerph21121638_

Round 1

Reviewer 1 Report

Comments and Suggestions for Authors

The paper analyzed the impacts of Low-Traffic Neighbourhoods (LTN) in reducing air pollution and the local community's perception of its impacts on health and well-being. The findings reveal a reduction in air pollution levels but with a polarizing overall scheme acceptance and spatial distribution of LTN's benefits. 

In general, the paper is well-written with a comprehensive presentation of the research background, detailed methodology, and well-discussed results. The topic itself is very relevant to addressing human and environmental health concerns. Reports, challenges, success stories, and lessons from this type of initiative applied in low- to zero-emission zones in most developed countries can be emulated in developing and other developed countries.

There are only minor comments to further improve the manuscript.

1. The originality of the paper can further be highlighted by explicitly identifying the research gap and the proposed academic contribution of the paper.

2. The methodology includes a survey of the perception of the community. Therefore, the study should describe the Ethical Considerations in research involving human participants (and a vulnerable group: 65+). If possible/available, present the Research Ethics Board Approval and other relevant information (see author guidelines)

3. Justify the type of sampling done in this study and how you arrived at 212 respondents (reduced to 210). Did you calculate the sampling size or base the number on data saturation?

4. Describe how you checked the validity and reliability of the research instrument (online survey).

5. The significance of the study can be improved by providing broader implications in the Discussion (or Conclusion) such as relevant policies on improving human/environmental health through this type of initiative, which can also be applied in other (particularly developing) countries. 

6. The quality of the figures must be improved by ensuring all fonts are readable (e.g. Figs. 1-3 are not readable). Also, make all fonts uniform for all figures, you may use the journal font (Palatino Linotype). 

7. Proofread the paper for minor errors. e.g. quotation mark in the Title

8. Make all units and decimal places uniform. e.g. µg/m³ vs µg /m³; 12.0% vs 12%; 40µg/m³ vs 40 µg/m³ 

9. Use uniform English. e.g. behaviour vs behavior

10. Please check the unit of CO2 (pollutant). It should be mg/m³ and not µg/m³ 

Reviewer 2 Report

Comments and Suggestions for Authors

Dear Authors,

Thank you for the opportunity to review your manuscript, which investigates the impacts of the Low-Traffic Neighbourhood (LTN) in Kings Heath, Birmingham, UK. The study aims to evaluate the scheme's effectiveness in reducing air pollution and explores local community perspectives on its perceived impacts on health and well-being. This is a valuable and timely study for the JERPH, but several areas need attention before it can be considered for publication.

Please find my detailed comments below:

Introduction:

The introduction would benefit from a clearer focus on Birmingham, UK, with a specific emphasis on how car traffic impacts air pollution in this city. This will help frame the study within the local context more effectively.

Aim of the Study:

Providing additional justification for selecting Birmingham, UK, as the case study could strengthen this section. An elaboration on why this location is particularly relevant would add depth.

Page 4 of 22 (Lines 164-180):

Further information on Birmingham’s city profile—such as urban form, population density, building density, GDP, income per capita, public transportation network, and social and economic background—is recommended. These factors are significant as they influence car use and, consequently, air pollution levels.

3.1 Air Quality Monitoring:

Including a map that shows the air quality monitoring sites would enhance clarity and allow readers to better understand the spatial coverage of the study.

3.2 Survey Questionnaire:

Details regarding the survey questionnaire used in this study are currently insufficient. Please provide a more thorough explanation of how the questionnaire was designed or developed. Additionally, the study does not appear to address the questionnaire's reliability and validity, which is essential to ensure data robustness.

3.2.3 Data Analysis:

The current analysis appears too simplistic for a publication in an indexed journal like IJERPH. A stronger justification for the analytical methods chosen is needed to demonstrate that they are both appropriate and rigorous.

Discussion:

The research findings are well connected to the literature, and this section does not appear to require significant changes.

Conclusion:

This section is acceptable; however, it could be strengthened by offering a more comprehensive conclusion that highlights the study’s implications and generalizations for the current body of knowledge. Suggestions for future research would also add value.

Thank you again for your submission. I look forward to seeing the revised manuscript.

Best regards,

Round 2

Reviewer 2 Report

Comments and Suggestions for Authors

Dear Authors,

Thank you for your efforts in revising the manuscript. I appreciate the improvements made, and it now meets most of the requirements for publication.

However, I would like to highlight a point regarding the research method section. While the employed method aligns well with the research objectives and is appropriate for this study, I kindly request that you elaborate further on the eligibility and capacity of the mixed-method approach.

This will enhance readers' understanding and awareness of how it supports achieving the study’s objectives.

Thank you for your attention to this matter.

Bests,

Author Response

Dear Authors,

Thank you for your efforts in revising the manuscript. I appreciate the improvements made, and it now meets most of the requirements for publication. However, I would like to highlight a point regarding the research method section. While the employed method aligns well with the research objectives and is appropriate for this study, I kindly request that you elaborate further on the eligibility and capacity of the mixed-method approach. This will enhance readers' understanding and awareness of how it supports achieving the study’s objectives.

Response - An introductory paragraph has been added the the Materials and methods section (LL188-200) to provide more details about the mixed method approach used in the study and the rationale for employing such approach. The following text was included in the text: “The study employs a convergent parallel mixed method approach using quantita-tive and qualitative data to investigate the impacts of the LTN on Kings Heath and the community’s perspective on the scheme. Data collection methods employed for this study comprise air quality monitoring sensors (quantitative) and an online survey questionnaire (quantitative and qualitative). The convergent parallel design approach involves collecting qualitative and qualitative data at the same time, analysing them independently, and then exploring the results to determine whether the findings sup-port or contradict each other [53]. The convergent parallel design approach was employed to provide a more comprehensive understanding of the impacts of the LTN, collecting richer data from both numerical data and contextual participants’ narratives related to opinions, perceptions and potential resistances about the scheme. This approach allowed for triangulation between qualitative and quantitative data, with con-sequent enhanced analytical interpretation.”